# Asymmetric one-pot transformation of isoflavones to pterocarpans and its application in phytoalexin synthesis

Philipp Ciesielski[1] & Peter Metz [1✉]

Phytoalexins have attracted much attention due to their health-promoting effects and their vital role in plant health during the last years. Especially the 6a-hydroxypterocarpans glyceollin I and glyceollin II, which may be isolated from stressed soy plants, possess a broad spectrum of bioactivities such as anticancer activity and beneficial contributions against western diseases by anti-oxidative and anti-cholesterolemic effects. Aiming for a catalytic asymmetric access to these natural products, we establish the asymmetric syntheses of the natural isoflavonoids (−)-variabilin, (−)-homopterocarpin, (−)-medicarpin, (−)-3,9-dihydroxypterocarpan, and (−)-vestitol by means of an asymmetric transfer hydrogenation (ATH) reaction. We successfully adapt this pathway to the first catalytic asymmetric total synthesis of (−)-glyceollin I and (−)-glyceollin II. This eight-step synthesis features an efficient one-pot transformation of a 2'-hydroxyl-substituted isoflavone to a virtually enantiopure pterocarpan by means of an ATH and a regioselective benzylic oxidation under aerobic conditions to afford the susceptible 6a-hydroxypterocarpan skeleton.

---

[1] Fakultät Chemie und Lebensmittelchemie, Professur für Organische Chemie I, Technische Universität Dresden, Bergstraße 66, 01069 Dresden, Germany.
✉email: peter.metz@chemie.tu-dresden.de

**D**ue to the diverse pharmacological properties, the pterocarpan [syllable division ptero-carpan] scaffold is an interesting target for research[1–4]. The 6a-hydroxypterocarpans (−)-glyceollin I (**1**) and (−)-glyceollin II (**2**) are phytoalexins isolated from stressed soybeans (Fig. 1). Klarman and Sanford were the first to isolate a single compound with antifungal properties from soybeans in 1968, and the correct structures of **1** and **2** were assigned in the 1970s[5–7]. In soy plants, the glyceollins are synthesised de novo from the isoflavone daidzein in response to pathogen attack such as fungi, bacteria, UV irradiation, or abiotic elicitors such as silver nitrate or aluminium chloride[8–11]. **1** and **2** belong to a group of at least eight A-ring prenylated 6a-hydroxypterocarpans, which are referred to as glyceollins[12]. In addition to the antifungal, antibacterial, and anti-nematode activities, which are important for the soy plant itself, the glyceollins have attracted much attention recently due to their activity as selective oestrogen receptor modulators and their anti-inflammatory, antiproliferative, anti-oxidative, and anti-cholesterolemic activities[13,14]. The antitumour activity of **1** is not limited to oestrogen-positive cancers such as mammary carcinoma or ovarian cancer, as it exhibits an antiangiogenic and anti-vasculogenic activity, too[15,16], as well as other non-oestrogen related anticancer mechanisms[13].

Due to the interesting bioactivity of glyceollins and the difficulties to isolate them in pure form from soy, the asymmetric total synthesis of these compounds became desirable. In 2008, the first total synthesis of (−)-glyceollin I (**1**), which also yielded minor amounts of (−)-glyceollin II (**2**), was published by the group of Erhardt and scaled up in 2011 to a multigram synthesis[17–20]. These studies utilised a Sharpless asymmetric dihydroxylation of an isoflav-3-ene with stoichiometric amounts of osmium tetroxide and the chiral ligand[21]. In 2014, Kohno et al. published an alternative approach to the isoflav-3-ene intermediate of the Erhardt synthesis[22]. Malik et al. also utilised a benzoin condensation for the synthesis of racemic glyceollin II (**2**)[23]. In our research group, a strategy relying on the epoxidation of an isoflav-3-ene led to racemic **1** and **2**, only[24].

The phytoalexin (−)-variabilin (**3**)[25,26] is a natural product with antifungal properties, which promotes the differentiation of neural stem cells, and influences the reabsorption of glucose in the kidneys[27–29]. Possessing the susceptible 6a-hydroxypterocarpan skeleton it represents a suitable model structure to establish a synthetic pathway toward the glyceollins I (**1**) and II (**2**). Our previous strategy to this 6a-hydroxypterocarpan via epoxidation of a 2H-chromene involved several unstable intermediates ruling out an upscaling of the synthesis[30], and thus encouraged us to attempt a different approach, which also enabled the synthesis of the phytoalexins (−)-homopterocarpin (**4**), (−)-medicarpin (**5**)[31–34], (−)-3,9-dihydroxypterocarpan (**6**)[35], and (−)-vestitol (**7**)[33,36].

Here, we communicate the asymmetric syntheses of the natural isoflavonoids **1**–**7** by means of an asymmetric transfer hydrogenation (ATH).

## Results

**The synthetic plan.** As one of the key transformations of our approach, we envisioned the introduction of the sensitive tertiary alcohol functionality of variabilin (**3**) by a regioselective biomimetic oxidation at the benzylic carbon atom 6a of the pterocarpan **4** (Fig. 2a). Pterocarpan **4**, obtained from **5** by simple methylation, might be accessible by acid-catalysed cyclisation of an isoflavan-4-ol after debenzylation of **8**[21]. We aimed to form **8** by ATH with dynamic kinetic resolution from the racemic 2′-hydroxyisoflavanone **9**. The ATH has proved to be an exceptionally powerful tool to establish the two consecutive stereogenic centres of isoflavan-4-ols from isoflavanones[37], but 2′-substituted isoflavanones proved to be less reactive under these conditions[38]. Racemic **9** might be accessible via 1,4-reduction of the corresponding isoflavone **10**. Isoflavones with a wide range of substituents are readily prepared by Suzuki coupling of iodochromones and phenylboronic acids[39].

Based on the successful execution of this synthetic plan for variabilin (**3**), we traced back the glyceollins I (**1**) and II (**2**) to the common intermediate **11** by pyran ring annellation and subsequent desilylation (Fig. 2b). The 6a-hydroxy group might be established by benzylic oxidation, again. At this point, we took up the challenge to use the 2′-hydroxyl-substituted isoflavone **13** as the substrate for an ATH cascade with dynamic kinetic resolution to give the desired pterocarpan **12** in a single transformation. The requisite isoflavone may be prepared rapidly from phenylacetic acid **14** by the reported Friedel–Crafts acylation of resorcinol and subsequent cyclisation to generate the isoflavone skeleton[40], demethylation, and chemoselective propargylation.

**First-generation synthesis of methylated phytoalexins.** Isoflavone **17** was prepared by Suzuki coupling under the conditions of Magnus et al. (Fig. 3)[41], starting from the known iodochromone **15**[24], readily available over three steps in 94% yield from 2′,4′-dihydroxyacetophenone (for optimised conditions, see Supplementary Note 2), and the boronic acid **16**, accessible over three steps from 3-methoxyphenol (see Supplementary Note 2). 1,4-Reduction of the isoflavone **17** to the corresponding isoflavanone **18** was achieved with L-selectride[37]. Chemoselective removal of the MOM group in the presence of the surprisingly sensitive benzyl ether was successful under carefully tuned conditions in 74% yield, and thus, provided the substrate **9** for the ATH.

The ATH led to the isolation of isoflavan-4-ol **8** in 89% yield and excellent enantiomeric excess after optimisation of our previously reported conditions[37]. The 2′-hydroxyl-substituted substrate **8** proved to be even more reactive than 2′-unsubstituted ones and encouraged us to subject the isoflavone **10** to the established protocol, too. The substrate **10** was prepared by Suzuki coupling of iodochromone **15** with the boronic acid **19** (for preparation, see Supplementary Note 6). Quenching the

R[1] = R[2] = Me, R[3] = OH, (−)-Variabilin (**3**)
R[1] = R[2] = Me, R[3] = H, (−)-Homopterocarpin (**4**)
R[1] = R[3] = H, R[2] = Me, (−)-Medicarpin (**5**)
R[1] = R[2] = R[3] = H, (−)-3,9-Dihydroxypterocarpan (**6**)

(−)-Glyceollin I (**1**)    (−)-Glyceollin II (**2**)    (−)-Vestitol (**7**)

**Fig. 1 Examples of isoflavonoids.** Bioactive natural products with 6a-hydroxypterocarpan skeleton (**1**–**3**) and selected other phytoalexins.

**Fig. 2 Retrosynthetic analysis of 6a-hydroxypterocarpans. a** Synthesis of (−)-variabilin (**3**) via biomimetic benzylic oxidation of (−)-homopterocarpin (**4**). The pterocarpan skeleton is traced back to isoflavan-4-ol **8** which is generated by asymmetric transfer hydrogenation (ATH). **b** Synthesis of (−)-glyceollin I (**1**) and (−)-glyceollin II (**2**) via a one-pot transformation from isoflavone **13** to pterocarpan **12** involving an ATH cascade and acid-catalysed cyclisation.

**Fig. 3 Synthesis of isoflavanol 8.** Four-step approach to **8** via ATH of 2′-hydroxyisoflavanone **9** and optimised two-step synthesis of **8** via ATH of isoflavone **10**.

cross-coupling reaction with concentrated hydrochloric acid provided isoflavone **10** in a single step in 78% yield after crystallisation. The ATH cascade from **10** to isoflavan-4-ol **8** proceeded with a high yield of 92% and excellent enantioselectivity (>99% *ee*), even with only 1 mol% of chiral catalyst instead of 5 mol%. We envision, that a first hydride transfer generates a racemic isoflavanone. Initially, we thought that the intramolecular hydrogen bond between the 2′-hydroxyl group and the ketone causes a stronger polarisation of the enone and hereby facilitates the 1,4-reduction, as 2′-unsubstituted isoflavones were earlier shown to be unsuitable substrates under our ATH conditions[42]. However, several experiments with 2′-unsubstituted isoflavones in the presence of externally added hydrogen bond donors did

not support this theory. Possibly, the intramolecular hydrogen bond in **10** fixes the orientation of the aromatic B ring to minimise unfavourable steric interactions between substrate and catalyst. The *R* enantiomer of the intermediate isoflavanone is preferentially reduced in the crucial ATH step to give the isoflavan-4-ol **8** via the proposed transition state **TS 1**, while the *S* enantiomer racemises rapidly under the reaction conditions via keto–enol tautomerisation. The attractive CH–π interaction between the *p*-cymene ligand and the aromatic A ring of the substrate together with the avoidance of a destabilising steric interaction between the tosyl group and the aryl substituent at the heterocyclic C ring is supposed to be responsible for the high enantioselectivity and diastereoselectivity of this reduction[37].

**Fig. 4 Final steps of the synthesis of several natural isoflavonoids.** Reduction to (−)-vestitol (**7**), construction of the pterocarpan skeleton, and methylation to (−)-homopterocarpin (**4**).

Having established an efficient access to isoflavanol **8**, we headed for the cyclisation under the reductive conditions of Luniwal et al. (Fig. 4)[18]. In our hands, treatment of **8** with hydrogen gas and palladium on charcoal led to a mixture of the envisioned (−)-medicarpin (**5**) and the further reduced isoflavan (−)-vestitol (**7**) in varying amounts. Prolonged exposure to hydrogen gas gave (−)-vestitol (**7**), only, in 77% yield and with a slight drop of enantiomeric excess (97% *ee*), indicating that the second reduction might take place during the cyclisation at the stage of the transient *p*-quinone methide or after the cyclisation. Addition of a small excess of pyridine successfully inhibited the cyclisation, and the isoflavanol **20** was isolated in 96% yield (see Supplementary Note 9). Direct treatment of the reaction mixture with hydrochloric acid after completed debenzylation gave the cyclised pterocarpan **5** in 90% yield without erosion of the enantiomeric excess as a single product. Methylation of (−)-medicarpin (**5**) under common conditions gave rise to the phytoalexin (−)-homopterocarpin (**4**) in 96% yield[34].

In order to find suitable conditions for the biomimetic benzylic oxidation of **4**, we screened a broad range of methodologies (Table 1). No conversion was observed using lead(IV) acetate as described by Khan (entry 1), and no product was obtained with potassium permanganate as oxidant under various conditions (entries 2–4)[43–46]. The ruthenium-catalysed procedure of Du Bois produced the desired product, however only in low yield and purity (entry 5), as did the uncatalysed version with ammonium cerium(IV) nitrate (CAN) (entry 6)[47,48]. Finally, an aerobic oxidation of (−)-homopterocarpin (**4**) to (−)-variabilin (**3**) was achieved with good regioselectivity under conditions adapted from Yoshino et al. (entries 7–8)[49]. The phthalimide-*N*-oxyl radical generated from cobalt(II) acetate and molecular oxygen can abstract a hydrogen atom from either C-6a or C-11a. Combination of the benzylic radicals with oxygen and subsequent hydrogen abstraction from *N*-hydroxyphthalimide forms two regioisomeric hydroperoxides, which can be reduced with triphenylphosphine. It was crucial to perform the reaction at 0 °C in order to stabilise the intermediate hydroperoxides and avoid overoxidation of the substrate, even though the reaction took a whole day to reach complete conversion. (−)-Variabilin (**3**) was isolated in a good yield of 64% with >99% *ee*, while the minor product (+)-isosativanone (**21**)[50] was slightly racemised.

**Total synthesis of glyceollin I (1) and glyceollin II (2).** The isoflavone skeleton was built by Friedel–Crafts acylation of resorcinol with phenylacetic acid **14** in boron trifluoride diethyl etherate and subsequent cyclisation using *N*,*N*-dimethylformamide (DMF) as C₁ fragment following a known literature protocol (Fig. 5)[40]. Double demethylation under standard conditions yielded the isoflavone 2′-hydroxydaidzein (**23**)[51] in high yield and

purity by simple crystallisation. The chemoselective propargylation of **23** utilising the copper(I)-catalysed conditions from Bell et al. took place exclusively at the most acidic 7-hydroxy group, forming the propargyl ether **13** in a good yield of 83%[52]. An alternative preparation of isoflavone **13** via Suzuki coupling of an iodochromone with an appropriate boronic acid was also investigated (see Supplementary Note 18), but found to be inferior to the streamlined protocol depicted in Fig. 5.

The isolation of the product of the ATH cascade, the isoflavanol, was tedious, so we developed a one-pot transformation of isoflavone **13** to pterocarpan **24**, which proceeded with a high yield of 82% and excellent enantioselectivity (>99% *ee*). After completion of the transformation of isoflavone **13** to the isoflavanol overnight, the reaction mixture was diluted with ethanol and acidified with concentrated aqueous hydrochloric acid to afford pterocarpan **24**. Addition of ethanol to the reaction mixture proved to be essential to trigger the cyclisation under mild conditions at ambient temperature and within a short reaction time of about 20 min. In contrast, cyclisation in ethyl acetate was not complete after 3 h at 45 °C and led to multiple side reactions. Interestingly, when the ATH reaction was run at room temperature and worked up with hydrochloric acid, the pterocarpan **24** was isolated in 39% yield along with 42% of the racemic isoflavanone, the product of the first reduction step. Subsequent TIPS protection of **24** generated the silyl ether **12** almost quantitatively. The benzylic oxidation of **12** under the conditions established above at 0 °C took several days to reach complete conversion. The desired 6a-hydroxylated pterocarpan **11** was isolated as the major product in 58% yield next to 16% of the isoflavanone **25** resulting from hydrogen abstraction at C-11a. To our delight, we discovered that in the presence of three equivalents of hexafluoroisopropan-2-ol (HFIP)[53], the reaction could be performed at room temperature in a short reaction time of 1 h to afford **11** in 55% yield next to 19% of **25**. The byproduct **25** might be resubmitted to ATH and subsequent cyclisation to improve the overall efficiency of the synthesis.

Cyclisation of the propargyl ether **11** to form the pyran ring led to the two regioisomeric compounds **26** and **27** with predominant formation of **26** (Fig. 6). Heating in dimethyl sulfoxide (DMSO) produced **26** in somewhat lower yield than in *N*,*N*-dimethylformamide (DMF) due to minor loss of product during aqueous workup in order to remove DMSO completely. On the other hand, running the reaction in DMSO is simple, while performing the reaction in DMF requires a sealed tube. Alternatively, the pyran ring was readily constructed by gold(I)-catalysed hydroarylation of the terminal alkyne in 1,2-dichloroethane (DCE), too[54,55]. The anticipated inversion of regioselectivity due to steric hindrance[55] was overridden by the electronic situation. However, the production of 23% **27** along with 65% **26** enabled a more

## Table 1 Benzylic oxidation of (−)-homopterocarpin (4).

| Entry | Conditions[a] | | Yield 3[b] | Yield 21[b] |
|---|---|---|---|---|
| 1 | Pb(OAc)$_4$, | DCM/HOAc | 0% | 0% |
| 2 | KMnO$_4$, | Amberlite CG-120, $^t$BuOH/DCM/H$_2$O | 0% | 0% |
| 3 | KMnO$_4$, | NEt$_3$, H$_2$O/CHCl$_3$, H$_2$SO$_4$ | 0% | 0% |
| 4 | KMnO$_4$, | 0.1 eq. Bu$_4$NHSO$_4$, KOH, DCM/H$_2$O | 0% | 0% |
| 5 | CAN, | (Me$_3$tacn)RuCl$_3$·H$_2$O (0.1 equiv.), AgClO$_4$ (0.4 equiv.), $^t$BuOH/H$_2$O 6:1, 30 °C | ≈15% | 0% |
| 6 | CAN, | MeCN/H$_2$O 1:1 | ≈16% | 0% |
| 7 | O$_2$, | NHPI (0.2 equiv.), Co(OAc)$_2$ (0.05 equiv.), MeCN | 22% | –[c] |
| 8 | O$_2$, | NHPI (1.0 equiv.), Co(OAc)$_2$ (0.5 equiv.), MeCN, 0 °C; then PPh$_3$, Ar | 64% | 10% |
| | | | >99% ee[d] | 91% ee[d] |

[a]Reactions were performed at room temperature if not stated otherwise.
[b]Isolated yield.
[c]Not determined.
[d]Determined by chiral HPLC.

**Fig. 5 Application of the ATH in the syntheses of glyceollin I (1) and glyceollin II (2).** Isoflavone **13** was prepared over three steps and subjected to the ATH cascade and subsequent acid-catalysed cyclisation. Benzylic oxidation yielded the 6a-hydroxypterocarpan **11**.

efficient access to glyceollin II (**2**) than the uncatalyzed approach. Final desilylation with tetrabutylammonium fluoride (TBAF) afforded the natural products **1** and **2** in high yields of 90% and 98%, respectively.

**Second-generation synthesis of variabilin (3).** Reconsidering our approach to variabilin (**3**) with the knowledge from the synthesis of glyceollin I (**1**), we envisioned to extend our methodology to isoflavones bearing an unprotected hydroxy group on the A ring (Fig. 7). After some tuning of the reaction conditions, we were able to convert 2′-hydroxdaidzein (**23**) to the naturally occurring pterocarpan **6** in 73% yield and high enantioselectivity (99% *ee*) using DMSO as the solvent. The intermediate isoflavanol **29**—like **23** and **6** an intermediate in the biosynthesis of glyceollins[11]—may be isolated without addition of hydrochloric

acid in 70% yield, too. Double methylation gave rise to (−)-homopterocarpin (**4**), which is oxidised to (−)-variabilin (**3**) to complete this concise synthesis.

## Discussion

Our strategy towards 6a-hydroxypterocarpans gives access to the natural products (−)-glyceollin I (**1**), (−)-glyceollin II (**2**), and (−)-variabilin (**3**). The glyceollins **1** and **2** were obtained in a short reaction sequence of eight steps from commercially available (2,4-dimethoxyphenyl)acetic acid (**14**) in virtually enantiopure form (*ee* >99%) with total yields of 15% and 4.9%, respectively. Flavonoid **3** was synthesised with a total yield of 27% over 5 steps from the same starting material **14**. Furthermore, our approach gave access to several other naturally occurring phytoalexins in an efficient manner and high enantiomeric purity:

**Fig. 6 Final steps of the glyceollin syntheses.** Pyran ring annellation and deprotection.

**Fig. 7 Second-generation synthesis of (−)-variabilin (3).** ATH cascade and cyclisation from the globally deprotected isoflavone **23** to give (−)-3,9-dihydroxypterocarpan (**6**), methylation to (−)-homopterocarpin (**4**), and conversion of **4** to (−)-variabilin (**3**).

(−)-3,9-dihydroxypterocarpan (**6**, 43% total yield, 3 steps, 99% *ee*), (−)-homopterocarpin (**4**, 4 steps, 42% total yield, 99% *ee*), (−)-vestitol (**7**, 6 steps, 52% total yield, 97% *ee*), and (−)-medicarpin (**5**, 6 steps, 61% total yield, >99% *ee*).

Moreover, the conversion of an isoflavone to a 6a-hydroxypterocarpan mimics successfully the biosynthesis of 6a-hydroxypterocarpans. This efficient route was enabled by three major accomplishments: a very short preparation of the isoflavones **13** and **23**, the development of the ATH from an isoflavone to the corresponding isoflavanol, and the identification of a suitable methodology for regioselective benzylic oxidation of the pterocarpan skeleton.

## Methods

**General information.** All reactions were performed under argon atmosphere if not stated otherwise. All other commercially available reagents were used as received without further purification. DMF and MeCN were dried over 4 Å molecular sieves. Dry DCM and THF were obtained from a solvent purification system MB-SPS-800. Et$_3$N was freshly distilled over CaH$_2$. TLC was performed on Merck silica gel 60 F$_{254}$ 0.2 mm precoated aluminium plates. Product spots were visualised by UV light (254 nm) and subsequently developed using anisaldehyde solution or permanganate solution as appropriate. Flash column chromatography was carried out using silica gel (Merck, particle size 40–63 μm). Melting points were measured on a Wagner & Munz PolyTherm A and are uncorrected. Infrared spectra were recorded on a Thermonicolet Avatar 360 instrument using ATR. NMR spectra were recorded on a Bruker AC-300 P (¹H: 300 MHz, ¹³C: 75 MHz), on a Bruker DRX 500 P (¹H: 500 MHz, ¹³C: 126 MHz) or a Bruker AC-600 P (¹H: 600 MHz, ¹³C: 151 MHz) spectrometer. Chemical shifts (δ) are quoted in parts per million (ppm) downfield of tetramethylsilane, using residual proton-containing solvent as internal standard (CDCl$_3$ at 7.27 ppm (¹H) and 77.00 ppm (¹³C) or MeOD-$d_4$ at 3.31 ppm (¹H) and 49.15 ppm (¹³C) or acetone-$d_6$ at 2.05 ppm (¹H) and 29.92 ppm (¹³C) or DMSO-$d_4$ at 2.50 ppm (¹H) and 39.51 ppm (¹³C). Mass spectra were recorded with an Agilent 5973N detector coupled with an Agilent 6890N GC (GC–MS, 70 eV) or else with a Bruker Esquire-LC (direct injection as a methanolic NH$_4$OAc solution, ESI). HRMS spectra were recorded on an Agilent 6500 Series Q-TOF (ESI-TOF). Optical rotations were measured on a Perkin Elmer 341 LC polarimeter. Elemental analysis was performed on a Hekatech EA 3000.

Enantiomeric excesses were determined by chiral HPLC on an Agilent 1100 Series with photodiode array detector (DAD) at ambient temperature.

Full experimental details, procedures, and characterisation for new compounds are included in the Supplementary Information.

**One-pot ATH cascade/cyclisation.** Preparation of the catalyst solution: [Ru(*p*-cym)Cl$_2$]$_2$ (3.7 mg, 6.0 μmol) and (*R,R*)-TsDPEN (5.0 mg, 13.3 μmol) were dissolved in EtOAc (0.36 mL) in a 5 mL round-bottomed flask. Another flask containing triethylamine (1.5 mL) was cooled to 0 °C, and formic acid (0.5 mL) was added. The mixture was stirred vigorously for 5 min at room temperature. An aliquot of this mixture (0.94 mL) was added to the ruthenium catalyst, and the resulting solution was stirred for another 5 min.

In a 5 mL round-bottomed flask isoflavone **13** (146.6 mg, 0.44 mmol, 1.0 equiv.) was suspended in EtOAc (0.54 mL) at 45 °C, and an aliquot of the catalyst solution (0.47 mL, ca. 1 mol% catalyst) was added. The reaction mixture was stirred at 45 °C for 19 h, cooled to room temperature and diluted with ethanol (4.4 mL). Hydrochloric acid (37%, 0.11 mL, ca. 3.0 equiv.) was added, the mixture was stirred for 13 min, and another portion of hydrochloric acid (37%, 0.11 mL, ca. 3.0 equiv.) was added. After 7 min, the reaction was quenched by addition of saturated aqueous NH$_4$Cl solution, and the mixture was extracted three times with EtOAc. The combined organic layers were dried over MgSO$_4$, the solvents were removed in vacuo, and the residue was purified by flash chromatography (DCM:EtOAc 2:1 *v/v*) to afford pterocarpan **24** (115.2 mg, 82%, >99% *ee*) as an off-white solid foam.

**Benzylic oxidation.** Pterocarpan **12** (125.9 mg, 0.26 mmol, >99% *ee*), *N*-hydroxyphthalimide (42.9 mg, 0.26 mmol, 1.0 equiv.) and cobalt(II) acetate (23.3 mg, 0.13 mmol, 0.5 equiv.) were dissolved in MeCN (5.3 mL) in a 10 mL round-bottomed flask, and 1,1,1,3,3,3-hexafluoropropan-2-ol (0.08 mL, 0.79 mmol, 3.0 equiv.) was added. The argon atmosphere was replaced with oxygen (balloon), and the reaction mixture was stirred for 1 h at room temperature. The oxygen atmosphere was replaced with argon, and triphenylphosphine (69.0 mg, 0.26 mmol, 1.0 equiv.) was added. After additional stirring for 45 min, the reaction was quenched by addition of saturated aqueous NH$_4$Cl solution. The mixture was extracted three times with EtOAc, the combined organic layers were dried over MgSO$_4$, the solvents were removed in vacuo, and the residue was purified by flash chromatography (DCM:EtOAc 30:1 *v/v*) to afford 6a-hydroxypterocarpan **11** (72.2 mg, 55%, >99% *ee*) as a brown gum and isoflavanone **25** (24.6 mg, 19%, 92% *ee*) as a yellowish solid.

## Data availability
The authors declare that the data supporting the findings of this study are available within the paper and its Supplementary Information file.

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

## Acknowledgements

We thank Dr. Tao Qin (TU Dresden) for first experiments on the ATH of isoflavones and Dr. Tilo Lübken (TU Dresden) and Dr. Ingmar Bauer (TU Dresden) for analytical support. Support by the Open Access Publication Funds of the SLUB/TU Dresden is gratefully acknowledged.

## Author contributions

P.M. conceived, designed, and directed the project, and P.C. designed and performed the experiments.

## Competing interests

The authors declare no competing interests.
