## [Peer Review File · Nature Communications]

Reviewer #1 (Remarks to the Author):

his manuscript describes a catalytic asymmetric access to pterocarpan and related several bioactive natural products in an efficient and divergent manner. In many transformations, high enantioselectivities could be obtained, especially the highly efficient construction of pterocarpan skeleton containing two vicinal stereogenic centers from isoflavone in one pot. This key asymmetric transfer hydrogenation reaction was developed by the same group. Hence, the reviewer recommends this manuscript to be published in Nature Communications once the following points listed below are addressed.

1. Since the transformation of isoflavone 13 to the isoflavanol 24 is the key step. The author also described the second reduction of the ATH cascade as the rate determining step. The proposal mechanism or more graphical explanation was suggested.
2. In the biomimetic benzylic oxidation step of the pterocarpan skeleton, the major byproduct was isoflavanone. Did the author try to transfer it to isoflavanol by ATH reaction and reuse it after cyclization? Then the whole synthesis efficiency could be improved.

Reviewer #2 (Remarks to the Author):

I have read "Development of an asymmetric one-pot transformation of isoflavones to pterocarpan and its application in phytoalexin synthesis", and I've synthesized several pterocarpan, so its fair to say that I have a little knowledge of this area.

Given the publication of the "Enantioselective Synthesis of Isoflavanones by Catalytic Dynamic Kinetic Resolution" by the same PI, Org Let, 2017, 19, 11, 2981, I see no urgency for publications of this manuscript as a communication. Additional comments are below.

Most, if not all of the transformations in this strategy are fairly well-known. A quick search shows over 104 REAxyS example of the cyclization to the tetracyclic compound. In my opinion, there is not obvious synthetic efficiency gains relative to other strategies, which deliver these target compounds in comparable number of steps and with higher ee to warrant publication as a communication based upon strategy. It also appears to me that there are several steps, which are not clearly delineated over arrows, which increases the appearance of overall efficiency of their strategy. Lastly, the source of SM 15 was not specified, and appears to take three steps to make. I do believe that the oxidation of 29 to the benzylic tertiary compound 3, is somewhat uncommon, but it uses Ishii conditions.

Is this solid work (Yes), should it be published somewhere (yes), Is this a great application of the PI's chemistry (Yes). However, I'd recommend publishing elsewhere such as Organic Letters, or JOC, or Angew Chemie.

Reviewer #3 (Remarks to the Author):

This is a really impressive piece of synthetic chemistry, which pulls together several diverse synthetic transformations to create an extremely efficient and concise synthesis of several highly

challenging target molecules.

Pivotal to the success of the synthesis is the use of a highly selective asymmetric transfer hydrogenation reaction to achieve a dynamic kinetic resolution of a racemic precursor, using methodology developed by the PI and his team recently. The work is extended here however to a clever 1,4- reduction linked to the ATH/DKR process.

Other notable reactions include a Pd-catalysed coupling reaction to create the key reduction substrate, an efficient cyclisation onto a benzylic cation to create a ring, and a directed introduction of a hydroxy group using a Ru-catalysed oxidation.

Taken together these create an impressive route to the product.

The work has clearly been carried out to a very high standard, as evidenced by the extremely detailed and high quality supporting information. Compounds have been fully characterised and the results are supported by NMR and HPLC spectra.

Whilst the route and the chemistry are impressive, the ATH/DKR does closely relate to previous work by the group (references 37 and 38) and the same directing principles are applied here. Since the route revolved around this pivotal process (albeit with an added 1,4- reduction), this limits its novelty. The principle of the ATH/DKR of such compounds is relatively well-established now.

The rest of the synthesis is brilliantly conceived and executed, however the admittedly-challenging hydroxylation reaction is sub-optimal, with modest yield and significant byproduct which has to be removed. Likewise the cyclisation in Figure 6 gives a mixture of products. Fortunately, both of these are valuable synthetic precursors, however it would be a stronger synthesis if the reaction had been selective for one of these.

Overall, this is an impressive piece of chemistry, and for most journals a highly likely accept. However Nature Communications sets the highest possible standards of international science of broad interest, and a significant breakthrough is required to warrant publication, as well as broad interest. This paper will interest a wide range of scientists but most significantly those working on synthetic and medicinal chemistry only. Unfortunately, the methodology employed in the key step is so closely related to (excellent) previous work that it cannot be regarded as having the required novelty or advancement that this journal requires.

Therefore I will unfortunately not be able to recommend this paper for publication. However I am sure that it will easily get into any one of a number of high quality international journals and encourage the authors to resubmit their work elsewhere soon.

Point-by-point response to the reviewers' comments

Reviewer #1:

1. Since the transformation of isoflavone 13 to the isoflavanol 24 is the key step. The author also described the second reduction of the ATH cascade as the rate determining step. The proposal mechanism or more graphical explanation was suggested.

The product distribution of the reaction at room temperature indicates the second reduction of the ATH cascade as the rate determining step, as no starting material was reisolated. However, since the mechanism has not been proofed so far, and no kinetic measurements were undertaken, we retract our statement about the rate determining step from the paper, as it only reflects our personal view on the reaction process.

2. In the biomimetic benzylic oxidation step of the pterocarpan skeleton, the major byproduct was isoflavanone. Did the author try to transfer it to isoflavanol by ATH reaction and reuse it after cyclization? Then the whole synthesis efficiency could be improved.

We did not try to reuse the isoflavanone 21 isolated after the benzylic oxidation, as it was obtained in 10% yield, only. While we did not try to reuse the isoflavanone 25, either, the idea to improve the overall efficiency of the synthesis by submitting it to the established reaction conditions for ATH and cyclisation is great. We strongly believe that it is possible to obtain the pterocarpan 12 or else the deprotected pterocarpan 24 using the conditions that we applied to isoflavone 13. Unfortunately, we are not able to conduct this transformation in our laboratories in the near future due to the COVID-19 pandemic (our university is operating in emergency mode). However, we appreciate this suggestion and added a comment on this matter to the manuscript.

Reviewer #3:

Since the route revolved around this pivotal process (albeit with an added 1,4- reduction), this limits its novelty. The principle of the ATH/DKR of such compounds is relatively well-established now.

The pronounced impact of the 2'-hydroxy group on the ATH reaction was not "well-established" at all. Indeed, its influence on the reaction was completely unprecedented and turned out to be crucial to this synthesis.

The rest of the synthesis is brilliantly conceived and executed, however the admittedly-challenging hydroxylation reaction is sub-optimal, with modest yield and significant byproduct which has to be removed.

Our results for the benzylic hydroxylation, which were obtained after extensive optimisation, are clearly the best for this transformation so far. Moreover, the ratio between product and byproduct is rather good (64:10) for substrate 4. For substrate 12 the ratio is not as high (55:19 to 58:16 - the latter ratio has now been added to the manuscript text) but as we mentioned above (response to reviewer #1) the byproduct might well be recycled.

Likewise the cyclisation in Figure 6 gives a mixture of products. Fortuitously, both of these are valuable synthetic precursors, however it would be a stronger synthesis if the reaction had been selective for one of these.

The selectivity of the cyclisation is indeed rather high (7:1) for the precursor 26 leading to glyceollin I, which is particularly important due to its interesting bioactivity.